# Acceptability of the Wulira app in assessing occupational hearing loss among workers in a steel and iron manufacturing industry

Immaculate Atukunda[1], Andrew Weil Semulimi[2,3], Festo Bwambale[2], Joab Mumbere[2], Nelson Twinamasiko[2], Mariam Nakabuye[2], John Mukisa[4], David Mukunya[5], Charles Batte[2,3]*

1 Department of Ophthalmology, School of Medicine, College of Health Sciences, Makerere University, Kampala, Uganda, 2 Department of Medicine, School of Medicine, College of Health Sciences, Makerere University, Kampala, Uganda, 3 Climate and Health Unit, Tree Adoption Uganda, Kampala, Uganda, 4 Department of Immunology and Molecular Biology, School of Biomedical Sciences, College of Health Sciences, Makerere University, Kampala, Uganda, 5 Faculty of Health Sciences, Department of Community and Public Health, Busitema University, Mbale, Uganda

* batchaux@gmail.com

## Abstract

### Background

Industrial workers are at a high risk of acquiring noise induced hearing loss, yet there is minimal hearing loss screening of such groups of people. Pure Tone Audiometry (PTA), the gold standard for hearing loss screening, is expensive, and not readily available at health sites. Mobile audiometry can bridge this gap. However, there is limited knowledge on its acceptability in low-income countries like Uganda. We aimed to assess the acceptability of using the Wulira App, a validated mobile phone app, in assessing hearing loss among industrial workers in Kampala.

### Methods

We carried out a qualitative study in a steel and iron manufacturing industry in Kampala, in April 2021. Four Focus group discussions (FGDs) with 8 participants per FGD, and 12 In-depth Interviews (IDI), were conducted on the industrial workers. The industrial workers were first tested for hearing loss, then enrolled for the FGDs and IDI. A semi-structured interview guide was used. Audio recordings were transcribed verbatim. Themes were derived using thematic content analysis, borrowing from Sekhon's model of Acceptability of Health Interventions.

### Results

Industrial workers found the Wulira App user friendly, cheap, time saving, and an effective hearing loss assessment tool. However, barriers such as lack of smart phones, difficulty in navigating the app, and fear of getting bad news hindered the App's acceptability, as a hearing assessment tool.

**Data Availability Statement:** Datasets used in the current study cannot be shared publicly because

they contain identifiable participants' information. Data are available on request for researchers who meet the criteria for access to confidential data from Makerere University School of Biomedical Sciences Research and Ethics Committee coordinator: irbbiomedicalsciences@gmail.com and the corresponding author.

**Funding:** This study was funded by the Government of Uganda through the Research and Innovation Fund Makerere University, Fund no. MAKRIF/ DVCFA/ 026/ 20. AWS and MN are research fellows of the MakNCD program supported by the Fogarty International Centre of the National Institutes of Health under Award Number D43TW011401. The content is solely the responsibility of the authors and does not necessarily represent the official views of the Funders. IA was the Research and Innovation Fund grant recipient. The funders had no role in study design, data collection and analysis, decision to publish, or preparation of the manuscript.

**Competing interests:** Dr. Charles Batte is part of the team that developed the Wulira App. Dr. Charles Batte and Dr. Andrew Weil Semulimi are directors at Wulira Health Limited that owns the Wulira App. The other authors have no conflict of interest to declare. This does not alter our adherence to PLOS ONE policies on sharing data and materials.

**Abbreviations:** PTA, Pure Tone Audiometry; WHO, World Health Organization; FGDs, Focus group discussions; IDI, In-depth interviews; SSA, Sub Saharan Africa.

## Conclusion

Hearing loss assessment using Wulira App was acceptable to the industry workers. There is need of informing industrial workers on the essence of carrying out regular hearing loss screening, such that barriers like fear of getting screened are overcome.

## Introduction

Hearing Loss is increasingly becoming a public health threat, and currently ranks as third among the non-fatal disabling conditions [1]. According to the World Health Organization (WHO), 466 million people suffer from disabling hearing loss, and a further projection of 630 million people by 2030 [2]. Hearing loss has been associated with deleterious consequences, including higher unemployment rates, poor health, social isolation, depression, dementia, and increased mortality [2–4].

Exposure to noise in recreation areas increases the risk of developing hearing loss by 7%, for every 5 years [5]. Noise induced hearing loss ranks among the leading causes of occupational illness among industrial workers [6,7] which subsequently results in unemployment [2,8]. Early detection and prevention of noise induced hearing loss is critical in addressing this occupational hazard.

Pure Tone Audiometry (PTA), which is the gold standard for hearing loss screening, is quite costly, requires a specialist audiologist to operate in a sound- proof room, and is not readily available at health (especially rural) sites in sub-Saharan Africa (SSA). On the other hand, Mobile Audiometry (hearing loss screening using mobile phones) has been shown to be as effective as PTA, and has great potential in improving access to hearing loss screening services [9–13]. This concept has been utilized in the general population to screen for hearing loss [9,10,14], thus providing evidence for its use in industrial workers.

One of the available mobile app-based hearing loss tools in sub-Saharan Africa is the Wulira app, which has been validated against the gold standard (Pure Tone Audiometry), and found to have a specificity of 93.2% (right ear, 95% CI (88.1–95.4%), 91.5% (left ear, 95%CI (87.2–94.7), sensitivity of 91.4% (right ear, 95% CI (88.9–93.5%), and 88.4% (left ear, 95% CI (85.6–80.9) [14]. This shows that the Wulira App can be used for hearing loss screening services, especially in settings with limited access to PTA [9,10,14]. Previous studies [12,15] have shown that effective utilization of technological health interventions is affected by their acceptance within the target group.

Although a study done in the United States of America, a high-income country, showed that mobile audiometry can easily be utilised among industrial workers [12], this has not been established in low-income countries such as Uganda, where phone ownership is still low [11].

In assessing acceptability of a particular health intervention, sekhon et al. suggested a theoretical framework with seven constructs [16], which we drew on, in guiding our presentation of study findings from a qualitative study. This was necessary, to understand the acceptability of mobile audiometry in assessing hearing loss among industrial workers using Wulira App.

As countries work towards Universal Health Coverage, prevention of disabilities should be at the centre of policy, and all programs. Evidence from our work will be able to guide policy in this regard. This study aimed at assessing acceptability of using Wulira App, in hearing loss screening among industrial workers.

## Methods

### Study design

We carried out an exploratory qualitative study, using Focus group discussions (FGDs) and in-depth interviews (IDI), with industrial workers at a steel and iron manufacturing industry.

### Study population

The study was carried out in a steel and iron manufacturing industry in Kampala, in April 2021. Kampala is the capital city of Uganda, with over 1.5 million people. As of 2011, the steel and iron manufacturing industry employed the largest number of people in Kampala district, which was close to 8,233 people [17]. In Kampala, there are 3 steel and iron manufacturing industries, each employing more than 200 people [18]. One industry was selected among the three, by purposive sampling. The selected industry employed over 1,100 staff on permanent basis, thus having more than enough participants for our study.

We enrolled workers permanently employed by the industry.

Face to face approach of the participants was done.

### Sample size

We did not decide in advance the sample size, but rather used the data saturation principle [19]. Thirty-two participants were recruited for the FGDs, of which 12 were invited for the IDI.

### Participant selection

Participants for the FGDs were purposively sampled from the industry workers' list, provided by the human resource manager, so that the industry's daily production is not affected by absence of the workers participating in the study. Four mixed FGDs were carried out, with each FGD having eight participants. Furthermore, 3 of the most active participants (those that appeared to have more to say, but were not given enough time in the FGD, as observed by the moderator) in each FGD were invited to participate in the In-depth Interviews. Participant recruitment was not sex specific. The inclusion criteria included; permanent staff employed by the industry, who were above 18 years, and had consented to take part in the study. We excluded participants who had established history of hearing loss.

### Data collection

Data was collected from a private room at the industry, to enable confidentiality and comfortability of the participants, while being engaged by a Ugandan male medical doctor, who was trained in qualitative research data collection. No third party, apart from the participants and data collection team, was allowed to be present. Each participant was first assessed for hearing loss using Wulira App, and then enrolled for the FGDs. The hearing assessment was done by a trained audiologist (FB), using Wulira App installed on the study tablet. The assessment results were not communicated to the participants prior to attending the FGD and IDI.

Each FGD lasted 60 minutes, and had the same moderator (AWS). Participants were on a round table, with the moderator seated among them. Before the start of each FGD, the moderator shared with participants the objectives of the study, and that the discussion will help in guiding the implementation of mobile audiometry in hearing assessment. Written informed consent was thereafter provided by the participants, and the moderator started the discussion.

A semi structured interview guide based on Sekhon's Model of Acceptability of Health Interventions [16] was used for data collection (S1 FGD guide tool). All FGDs were audio recorded, and field notes taken by MN.

During the FGDs, the moderator noted the participants who gave more information, who were then invited for the IDI. Three participants from each FGD were recruited for the IDI. AWS carried out all the IDI, with the note-taker being MN. All IDI were audio recorded, as well with each IDI, lasting a minimum of 45 minutes, within the recommended time of similar IDI [20]. No repeat interviews were done.

The investigators underwent intensive training, before enrollment of participants for the FGDs and IDI. FGDs were conducted, until data saturation was reached. A total of 4 FGDs were carried out, with each FGD containing 8 participants, and 12 in-depth interviews. No participant refused to participate. After data collection and transcription, no transcript was returned to study participants.

## Data analysis

All audio recordings were transcribed verbatim. The initial codebook was developed after carefully analyzing the text line-by-line of each transcript through coding, using de-identified respondent identification numbers. JM assigned codes to relevant segments of the text, and similar or related codes aggregated to form themes. Themes were derived using thematic content analysis, borrowing from Sekhon's Theoretical Framework of Acceptability [16]. Words, sentences or paragraphs that conveyed a similar message were grouped as meaning units, which were then condensed and labelled with a code. JM aggregated similar codes to form categories. Categories were made to be mutually exclusive whenever that was possible, and to include all the information related to the content area being discussed. Categories were further analyzed, to form sub themes, and themes from their latent meanings [21,22].

The FGD findings were ratified and triangulated, with themes obtained from IDIs. A narrative was generated from the dominant themes. Some quotes are used to represent the narrative. Nvivo software was used to organize the analysis. Participants did not provide feedback on the findings. Findings of this study are reported according to consolidated criteria for reporting qualitative studies (COREQ) [23], as shown in S2 File.

## Ethics statement

Ethical approval was granted by Makerere University School of Biomedical Sciences Institutional Review Board (SBS-862), and Uganda National Council of Science and Technology (HS1237ES). Each participant provided written informed consent.

## Results

In this qualitative study, the majority of the participants were males (65.6%), with a median age of 27 years. Half of the participants had their highest education level being secondary school, with a majority of them (75%) working as machine operators (**Table 1**).

### Acceptability of hearing assessments using Wulira app

The results from the IDI and FGDs revealed themes that are presented using Sekhon's framework, to understand the acceptability of the hearing assessment, using the Wulira app. Overall, the hearing assessment using Wulira app was acceptable to the industrial workers.

**Table 1. Socio-demographic characteristics of the participants.**

| Characteristics | N (%) |
|---|---|
| **Sex** | |
| Male | 21 (65.6) |
| Female | 11 (34.4) |
| **Age Median, IQR** | 27 (25–30) |
| **Level of Education** | |
| Primary | 02 (6.3) |
| Secondary | 16 (50) |
| University | 09 (28.1) |
| Others | 05 (15.6) |
| **Job Description** | |
| Management | 06 (18.8) |
| Medical staff | 02 (6.25) |
| Machine Operators | 24 (75) |
| **Duration of Work** | |
| Less than 6 months | 03 (9.4) |
| Between 6 and 12 months | 04 (12.5) |
| Between 12 and 24 months | 09 (28.1) |
| More than 24 months | 16 (50) |
| **Hearing loss** | **00** |

## Affective attitude

Affective attitude denotes how an individual feels (either negatively or positively) about the intervention.

User friendliness:

Participants found using the *Wulira* app easy for them. The lay out of the app, and the guidelines for use were straightforward and intuitive. One participant stated

> "*To me using the Wulira App, the process of testing my ears was very easy. It was clear what was expected of me, it went on well*" -(FGD 2)

Improved health seeking behavior:

Some of the participants in the in-depth interviews and FGDs said the use of the *Wulira* app was the first mobile based app they had encountered in hearing test.

> "It was my *first time to test my hearing. It was so sensitive, I got the sounds very well, it was comfortable during the testing. I recommend Wulira because it is the only gadget right now which I know and to me it was very sensitive that is number one.*" (Interviewee 6)

For the rest who had tested their hearing, they had done so in the company clinic, although they could not remember how regularly they went there. This is illustrated by the quote:

> "*I have done a test in our clinic they also used the resonance tool. They hit it and then brought it next to my ear to test whether you can hear the disappearing sound*" (FGD 4)

Fear of getting bad news from the hearing testing:

Fear of getting bad news among the participants was a major barrier for the acceptability of the hearing screening test. Many of the participants mentioned hesitancy in coming for the tests, despite working in high noise environments at the factory.

"*According to the life we live, we are in a noisy area full day. However, I was fearing to even come for this testing. You never know what result I would get.*" (Interviewee 10)

Another participant interestingly offered:

"*I have been fearing to attend to medical personnel within the company for some tests. So when this new App (Wulira) testing came in, this new activity of testing came in. Everybody is just eager*" (Interviewee 4)

## Self-efficacy and burden

The next two themes within the Sekhon et al model were self-efficacy and burden. Self-efficacy is an individual's confidence that they can perform the procedures of the intervention, while burden is how much effort an individual thinks is needed for the successful outcomes of the intervention, if they participate. For this analysis, the sub-themes for self-efficacy and burden constructs overlapped, and are reported jointly.

**Self-efficacy.**   Easy to access:

Certain respondents felt that the *Wulira* app was easy to obtain and use for their own personal hearing assessments. They highlighted that the fact that they had smart phones and some internet access, they could download it on to their devices.

"*I will manage to get the App and install it on my phone and then I will have to get headsets and I can do it at any time I want. I use it when am home and on different types of people.*" (FGD 4)

Willing to test on their own:

Other participants felt that they would be able to perform the tests on their own in the future, if given appropriate training.

"*. . .I can do the hearing testing myself if I am taught how to use it. I can do it as many times as possible since we work in noisy sections. There are departments where noise is too much whereby that kind of noise can affect my ear so I would prefer five to six times in a month.*" (FGD 2)

Lack of smart phones:

The Wulira app's compatibility with only smart phone or other smart devices, was seen as a major barrier for the performance of hearing testing at the home, or rural areas.

"*. . . probably since it was just an application on the phone, someone may need to use the app, yet they do not have a smart phone which prevent them from testing*" (Interviewee 8).

Technology literacy:

A few participants mentioned challenges in navigating and using the *Wulira app* to assess their hearing. They noted the need to have internet to download it, and an individual to train them on how to use the app. As one participant said:

"*I would not know where to go to download the app and* even I didn't know what to do like me to do the testing but someone came here and helped me do the testing for free *(Interviewee 6).*

**Burden.**   Time efficiency:

Maximizing the outputs at the factory is a key goal for any of the managers of a production facility. Therefore, any health procedure that does not take the participants away from their job for long rhymes does well with their values of effective use of time. The participants noted that the use of the Wulira app for the screening of the hearing loss fitted into their schedules, and they felt that they would easily be compensated for it on return to their job stations, as it was a faster procedure and led to little or no losses.

"*. . .we used it (Wulira app) with its less effects, and this being a production company, it does not lead to time wastage so I would recommend the company to always use the Wulira app*" *(Interviewee 11) it did not affect my work as much because it was around ten to thirteen minutes and after the testing, I was able to make it up for the lost time of thirteen minutes*" *(Interviewee 3)*

It is cheap:

While there were a few participants concerned about intervention costs, most appeared to feel that the intervention could bring cost savings. The cost of the procedure had prevented some of them from accessing hearing testing. They saw this an opportunity to test at a cheaper price. As one indicated:

"*I thought if you are going for the hearing test you have to go to the hospital, and I even thought it was costly. However, this exercise was somehow very cheap and easy to access because most of the people have smart phones. If it comes and people can download and use it, it's very good*" *(Interviewee 10)*

## Perceived effectiveness and intervention coherence

Sekhon et al define perceived effectiveness construct as the extent to which an intervention is expected to achieve its purpose [16]. On the other hand, intervention coherence is the extent to which participants understand the intervention, and how it works.

**Perceived effectiveness.**   Had desired hearing testing qualities:

Several participants recognized the ability of the app to assess their hearing testing. The various frequencies that were generated in the app gave a range of results, for both high and low frequencies, unlike other hearing testing gadgets they had encountered.

"*. . .I like the consistency of how results are being generated from the Wulira app. I could hear the low and high frequencies through the headsets.*" *-(Interviewee 2)*

It is better technology:

To some participants who had ever experienced hearing testing, using the tuning fork and pure tone audiometry, the incorporation of new smart phone app technology in the Wulira app in screening for hearing loss, was viewed as an improvement, that would lead to better results of the testing. This would be crucial in areas where many participants need to be assessed and treated, if need arises.

"*I think the Wulira App is a better technology because with the tongs, well I feel like even the frequencies in which the sounds come from the tongs it may differ depending on the strength at which it has been hit.*" (Interviewee 12)

In addition, other participants appreciated the information technology invested in the Wulira app, which was faster and equivalent to previous analogue testing tools, like the tuning fork.

"*Within the medical checkups, there is also testing for hearing capacity amongst our workers, and we basically have been using the analogue medical tools (not IT programmed like the Y shaped tuning fork), which were being handled by the medical team. But now having this tool which uses an App, it will ease the process of our medical checkups with our workers*"- (Interviewee 2)

Potential benefits of Wulira app hearing screening test:
Some of the participants cited that the hearing testing using the *Wulira* app would provide information on the hearing status of their work colleagues. This may have future positive effects of the work environment noise protections and work place health policies. This is illustrated by the quote:

"*We will get feedback on the hearing tests quickly about our work colleagues and I believe that it (Wulira app testing) will help us to assess whether we need to improve on the working conditions so that people don't get hearing loss.*" (Interviewee 5)

**Intervention coherence.** Testing procedure was comprehendible:
All the participants knew the details of how the intervention was performed. They described the use of headphones and different frequencies, sent by an assessor to their ears. After this, they made signals to show that they had heard the sounds.

"*They started by checking our ears with a bright light to see whether I have wax and they asked if I have any problem with our hearing, after they connected the phone on to the Wulira App headset and then I would give signals in case I heard any sound like give a signal to the person who was interviewing. He would detect whether we can hear the low, high or mild, any sound or the high-pitched ones. However low it would be he told me that I had to give him a signal.*" (FGD 3)

## Opportunity cost and ethicality

Opportunity cost is defined as the potential loss or gain from other alternatives, when a choice is made, while ethicality is the extent to which the intervention is considered as a good fit with their values for these two constructs. The sub-themes overlapped, and are presented together.

**Opportunity cost.** Willing to leave work for hearing assessment:
Participants were willing to give up time they would have used for other activities, so that they would participate in the intervention. To most of the participants (10/12 in depth interviews), they did not lose much through their participation in the hearing test assessment:

"*The exercise took me about 15 minutes. It did not affect my work schedule as I utilized the big lunch break since the procedure was not time consuming*" (Interviewee 9)

Similar findings were found in the FGDs as shown:

"*it (hearing assessment with Wulira app) took around four minutes which as just so small and it didn't affect my work schedule because my schedule always takes longer than that*"(FGD 1)

Lack of time to do testing:
Inadequate time to participate in the hearing was also a barrier to accepting the hearing assessment using the Wulira app, since they work in sections of the production plant, that keep running throughout the day.

"*. . .I work in the very production section, and it is challenging to leave the station at any time I am on duty.*" (Interviewee 11)

**Ethicality.**   The use of the Wulira app to address the screening needs for hearing loss among the workers was viewed as suitable for their work lifestyle.
Personal relevance:
In the FGDs, the participants felt that the hearing assessment by the Wulira app was an intervention they could benefit from. They felt that they would be able to know their hearing status more frequently. One participant mentioned that:

"*With the Wulira App I think I will be able to do frequent screening to determine my hearing performance because with this App I don't have to go and see a doctor. I can do it any time I feel my ears have a problem or my hearing is not okay, and I think it's necessary because hearing is a major sense to us as humans.*" (FGD 4)

Willingness to do more frequent hearing tests:
Based on their current personal experiences, the study participants were willing to have routine hearing assessments at their workplace. They felt that this regularity might help them realize any hearing problems earlier, and seek faster and appropriate management. However, the frequency varied between participants, with ranges from once a month, to once a year.

"*. . .I would advocate for it. I would advocate for frequent screening as long as the frequency for screening is determined, like if at all they have adjusted like maybe depending on our noise levels and environment, if at all we do like quarterly or maybe twice a year depending on the frequency of usage, I would advocate for the frequent screening*" (Interviewee 2)

Interestingly to other participants, the frequency of routine screening would even be better, if dropped to biweekly clinic assessments, as shown in this quote:

"*. . .because Wulira app wouldn't incur very many costs, that would at least be two weeks. . ..*" (Interviewee 11).

## Discussion

This current study showed that hearing assessment, using Wulira App is acceptable, in addition the Wulira App, is easy to use, and an effective hearing loss screening tool. However, challenges like technology illiteracy and lack of smart phones were mentioned, as the barriers that could hinder Wulira App usage.

A Study done by Sabur et al, revealed that Digital technologies introduction in medicine faces barriers like end user acceptance [24]. Contrary to that, our results show that Wulira App was acceptable to the industrial workers. A previous study has shown that Wulira App has a similar sensitivity and specificity, at detecting hearing loss, when compared to the conventional PTA [13], hence this could explain its acceptability. This current result shows that Wulira App is an effective hearing loss assessment tool. Audiometry access in Uganda being minimal, Wulira App gives the option of prevailing valid audiometer data.

Participants in this study revealed that Wulira app was easy to use, however, they had challenges in navigating and knowing how to use the app in assessing hearing loss. A Study performed by Zhenzhen et al, revealed that for promotion of mHealth interventions, they should be user friendly [25]. Other studies have shown that difficulty in use of mHealth technologies in terms of navigation and network reliability, compromised the efficiency and usability of the technologies [26–28]. Wulira App team took careful measures to overcome such challenges, like making the app accessible offline and online, and embedding directory messages on how to use the App.

Generally, most individuals do not desire to be informed that they are ill, and resent the idea that their work life and life-style will be endangered by their health state [29–31]. Similarly, participants in our study feared the prospect of getting bad news, hence limiting the App's acceptability. Hearing loss is strongly linked to depression [32,33]. With this result, there should be an increase in education regarding the importance of screening for hearing loss. The decreased ability to hear is annoying in and of itself [34]. With screening, hearing loss is detected early enough, and treated appropriately [34]. The focus of screening is to separate individuals who have a potential hearing disorder, from those who don't [35].

Our results revealed that Wulira App's compatibility with only smartphones will be a major barrier for individuals in rural areas, or with no smart devices. In Uganda, 70.9% of the individuals own a mobile phone, however, of these, 15.8% have a smartphone, with more urban individuals in possession of these phones [11]. Primary inhibitor to the possession of smart devices is affordability of these devices [11,36]. With rapid changes and fall in smart phone prices, affordability of such devices will soon be solved, thus increasing usage of smart phone-based interventions. The employers could as well provide test personal hearing on a stand-alone smartphone on which the application is installed under the supervision of a technology skilled person to assist individual persons.

One of the challenges to getting hearing care is the cost of equipment and services [37,38]. A study done by Hussein et al., revealed that traditional diagnostic audiometers are expensive [37]. Contrary to that, participants in our study revealed that Wulira App is an intervention that could bring cost savings, when accessing for hearing testing. This finding conforms to the need to integrate low cost interventions, in countries that have limited health budgets [38,39].

Participants reported that Wulira App was effective in assessing hearing loss of both low and high frequencies, as it would give a range of results for both high and low frequencies, which was not the case with other testing gadgets they had interfaced with. This acceptance shows that Wulira App may be acceptable in assessing hearing loss of both low and high frequencies. In addition, participants revealed that Wulira App was faster in assessing hearing loss, thus being a time saver.

## Limitation

Participants were initially screened for hearing loss using the Wulira App, before being recruited for the FGDs. Of those screened, none was found to be having hearing loss. This might have influenced the participants' optimism and perceptions, when being later asked to

describe their experience and their perceived acceptability of the App. However, this potential bias was reduced by making participants unaware of their results before the FGDs, and using open ended questions when conducting the FGDs.

## Conclusion

The study suggests that Wulira App could be an acceptable, cost-effective, easy to use, and time saving tool in hearing assessment. However, challenges such as technology illiteracy, lack of smart phones, and fear of getting to know their test results, could hinder uptake. There is need of educating industrial workers on the essence of carrying out regular hearing loss screening, such that barriers like fear of getting screened are overcome.

## Supporting information

**S1 File. FGD guide tool.**
(DOCX)

**S2 File. Consolidated criteria for reporting qualitative research check list.**
(DOCX)

## Acknowledgments

We appreciate the administration of the steel and iron industry, which permitted the study to be carried out in their facility and mobilising its workers to actively enrol into the study. We are further grateful to the participants who took part in the study.

## Author Contributions

**Conceptualization:** Immaculate Atukunda, Andrew Weil Semulimi, Charles Batte.

**Data curation:** Andrew Weil Semulimi, Festo Bwambale, Joab Mumbere, Nelson Twinamasiko, Mariam Nakabuye, John Mukisa, Charles Batte.

**Formal analysis:** John Mukisa, David Mukunya.

**Funding acquisition:** Immaculate Atukunda.

**Investigation:** Nelson Twinamasiko.

**Methodology:** Festo Bwambale, Joab Mumbere, Nelson Twinamasiko, John Mukisa.

**Project administration:** Mariam Nakabuye, Charles Batte.

**Software:** Festo Bwambale, Joab Mumbere.

**Supervision:** Charles Batte.

**Validation:** Andrew Weil Semulimi, Mariam Nakabuye.

**Visualization:** Joab Mumbere, David Mukunya.

**Writing – original draft:** Immaculate Atukunda, Andrew Weil Semulimi, Nelson Twinamasiko.

**Writing – review & editing:** Immaculate Atukunda, Andrew Weil Semulimi, John Mukisa, David Mukunya, Charles Batte.

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
