## [Decision Letter · Decision Letter 0]

13 Dec 2021

PONE-D-21-32084ACCEPTABILITY OF THE WULIRA APP IN ASSESSING OCCUPATIONAL HEARING LOSS AMONG INDUSTRIAL WORKERS.PLOS ONE

Dear Dr. twinamasiko,

Thank you for submitting your manuscript to PLOS ONE. After careful consideration, we feel that it has merit but does not fully meet PLOS ONE’s publication criteria as it currently stands. Therefore, we invite you to submit a revised version of the manuscript that addresses the points raised during the review process.

We look forward to receiving your revised manuscript.

Kind regards,

Mohammad Hossein Ebrahimi

Academic Editor

PLOS ONE

“AWS and MN are research fellows of the MakNCD program supported by the Fogarty International Centre of the National Institutes of Health under Award Number D43TW011401.”

“This study was funded by the Government of Uganda through the Research and Innovation Fund Makerere University, Fund no. MAKRIF/ DVCFA/ 026/ 20. The content is solely the responsibility of the authors and does not necessarily represent the official views of the Research and Innovation Fund.

IA was the grant recipient.

“I have read the journal's policy and the authors of this manuscript have the following competing interests: Dr. Charles Batte is part of the team that developed the Wulira App. Dr. Charles Batte and Dr. Andrew Weil Semulimi are directors at Wulira Health Limited that owns the Wulira App. The other authors have no conflict of interest to declare.”

Reviewers' comments:

<ul><li>**Comments to the Author**

1. Is the manuscript technically sound, and do the data support the conclusions?

Reviewer #1: Yes

Reviewer #2: Partly

Reviewer #3: Yes

2. Has the statistical analysis been performed appropriately and rigorously?

Reviewer #1: Yes

Reviewer #2: Yes

Reviewer #3: N/A

3. Have the authors made all data underlying the findings in their manuscript fully available?

Reviewer #1: Yes

Reviewer #2: No

Reviewer #3: No

4. Is the manuscript presented in an intelligible fashion and written in standard English?

Reviewer #1: Yes

Reviewer #2: Yes

Reviewer #3: Yes

5. Review Comments to the Author

Reviewer #1: Dear authors thank you for your research:

-I preferred to mention the detailed of the Sekhon’s framework of Acceptability tool of data collection in the method section.

-regarding the sample size: could you please give some information about data saturation principle.

2you mention many limitations for the study that affect the validity of your results; what did you conduct to decrease the effect of them on your study results?

and you didn't ; why?

Reviewer #2: General comments:

- Please avoid starting a sentence as "There is/was/are/were" in scientific writing.

- Do not use the full stop after the title of the section.

Abstract:

- Please say when and where the study was done.

- It is not clear which were the inclusion/exclusion criteria.

- It is not clear what was tested with the FGDs.

- The results are not specific (e.g., how many are considered friendly or cheap; what time-saving means, etc.)

Introduction:

- "has increased the risk of developing hearing loss" how considerable the risk is?

- Please also report the 95% confidence interval associated with Se and Sp.

- "Previous studies" to which populations were these previously study conducted?

- "phone ownership is still low due to the cost" how low and how much does it cost relative to the income of persons who work in industries where hearing loss occurs?

- "Literature on the acceptability of mobile audiometry for monitoring purposes in Africa and Uganda is scarce" Briefly present these studies and their results.

- It is not clear which is hearing loss among industrial workers in the target population.

- End this section with the aim of your study.

Methods:

- It is not clear when the study was conducted.

- "industrial workers at a steel and iron manufacturing industry" Are the steel and iron manufacturing industry the most prevalent industry where hearing loss can occur?

- Why was this particular city and industry was chosen?

- The inclusion criteria are not sufficiently presented (e.g., how many years were they working in that industry, had the family history of hearing loss? etc.).

- It is not clear how the participant was recruited and how the participants were decided.

- Define "most active participants".

- Is there any reason why the participants were not evaluated for hearing loss by gold standard procedure?

- It is not clear which questions were in the interviews.

- It is not clear if the participants used the App on their own phones or not.

- It is not clear if the participants were or not with phone/smartphone skiled.

Results:

- Put the results either in text or in Tables. Do not duplicate the results in text and tables/figures.

- Do not include in the body of the table the units of measurements (e.g. %).

- "The results from the IDI and FGDs revealed themes that are presented using Sekhon’s framework to understand the acceptability of the hearing assessment using the Wulira app" this information is duplicated; please delete it.

- It is not clear how many participants agreed with the features presented in Table 2 as facilitator/barrier.

Discussion:

- Do not duplicate information (the aim of the study is already known).

- It is useful to begin the discussion by brieﬂy summarizing the main ﬁndings and explore possible mechanisms or explanations for these ﬁndings.

- Emphasize the new and important aspects of your study and put your findings in the context of the totality of the relevant evidence. State the limitations of your study, and explore the implications of your ﬁndings for future research and for clinical practice or policy.

- "Our results revealed that Wulira App’s compatibility with only smartphones will be a major barrier for individuals in rural areas or with no smart devices" a possible solution to this barrier is that the employer to provide the possibility to test personal hearing on a stand-alone smartphone on which the application is installed and why not with a supervision of a technology skilled person to assist individual persons.

- The main limitation of your study is the absence of performance evaluation of the used App since the respondent was disease-free and no standard gold diagnosis was made.

Conclusions:

- Not all sentences presented here are conclusions (e.g., 453-457).

- Please provide the date for ethics approval.

Reviewer #3: The reviewed article presents a qualitative evaluation of the acceptability of the Wulira App in assessing hearing loss among industrial workers in Kampala, Uganda.

Overall, this qualitative investigation produced a useful insight into both facilitating and hindering factors experienced by a sample of industrial workers after their hearing has been assessed using the evaluated App.

The evaluation involved 4 focus group discussions (FGD) with 8 participants per FGD and 12 in-depth interviews (IDI), conducted after all industrial workers were first tested for hearing loss using the evaluated App.

None of the investigated workers has been found to be suffering from hearing loss.

In this reviewer’s opinion, this might have influenced the participants’ optimism and perceptions, when being later asked to describe their experience and their perceived acceptability of the App.

Unless the audiometric test results were not communicated to participants until after their participation in the FGDs and IDIs (an aspect that should be clearly mentioned in the article), this represents a limitation that should also be mentioned by the authors, and which should be addressed by future studies, by including a larger number of participants which may ensure not only the qualitative-data saturation (which dictated sample size in this study), but also a reasonable number of participants with hearing problems to be detected by the App (thereby reflecting the prevalence of hearing loss in the targeted population of industrial workers).

Despite the major conflict of interest that has been declared by 3 of the authors (who own, direct and/or have participated in the development of the evaluated App), other than this possibly important limitation of the study (which needs to be clarified or stated along with those already underlined in the Limitations section), the scientific approach appears to have been conducted using fairly objective methods (as described in the Methods part of the study and as reported in accordance to the COREQ guidelines).

Also, lacking other evidence, the authors needed to rely on their own previous works when making accuracy/ sensitivity/ specificity claims about the investigated App. Given the aforementioned conflicts of interest, encouraging future independent studies of diagnostic accuracy and acceptability of the evaluated App would be desirable.

Further improvements of the article manuscript may also be achieved by addressing the following issues:

Lines 73-75: please specify the gold standard against which the authors have determined the Se and Sp of the evaluated App in their previous work. Also, please specify the sample size and the 95% CI for Se and Sp.

Lines 129-130: please detail what “most active participants” means, since this may have been a highly subjective choice, leading to the selection of the most optimistic and favorably-oriented participants towards the evaluated App.

Lines 143-144: this reviewer suggests replacing “Before start of each FGD, moderator shared” with “Before the start of each FGD, the moderator shared”.

Line 150: this reviewer suggests replacing “moderator noted” with “the moderator noted”.

Line 172: this reviewer suggests replacing “FGD findings were ratified and triangulated themes got from IDI” with “The FGD findings were ratified and triangulated with themes got from IDI”.

Line 174: please specify which open code software was used to perform the analysis.

Line 190: this reviewer suggests replacing “majority” with “the majority”.

Line 192: this reviewer suggests replacing “majority” with “a majority of them”.

Line 233: this reviewer suggests replacing “working high noise environments” with “working in high noise environments”.

Line 235: there is an unlikely omission of a sense-giving verb in the verbatim transcription of Interviewee 10: “You never what result I would get.”

Line 290: this reviewer suggests replacing “prevented some of them for accessing” with “prevented some of them from accessing”.

Lines 382-383: this reviewer suggests replacing “ranges from everyone month to once a year” with “ranges from once a month to once a year”.

Line 398: this reviewer suggests replacing “Study done” with “A study done”.

Lines 401-402: Once again, please specify the gold standard against which the Se and Sp of the App and of PTA have been determined, in order to reach a conclusion of “similarity” between the App and conventional PTA. Also, even if Se and Sp were indeed similar between the App and conventional PTA, it is not clear how that would influence or even “explain its acceptability”.

Lines 406-407: this reviewer suggests replacing “Study done” with “A study performed”.

Line 420: this reviewer suggests replacing “to separate individuals with potential hearing disorder from those who don’t” with “to separate individuals who have a potential hearing disorder from those who don’t”.

Line 424: this reviewer suggests replacing “in possession with these phones” with “in possession of these phones”.

Line 435: please resolve the doubling in “would would”.

Lines 476-478 and 480: There are several abbreviations in the list which do not appear in the text of this article: YLD, OHL, NIHL, MDR-TB. Those abbreviations might have been useful in previous publications of the authors, but should be removed from this article.

6. PLOS authors have the option to publish the peer review history of their article (what does this mean?). If published, this will include your full peer review and any attached files.

**Do you want your identity to be public for this peer review?** For information about this choice, including consent withdrawal, please see our Privacy Policy.

Reviewer #1: No

Reviewer #2: No

Reviewer #3: No

</li></ul>

---

## [Author Response · Author response to Decision Letter 0]

8 Feb 2022

Reviewer #1: Dear authors thank you for your research:

-I preferred to mention the detailed of the Sekhon’s framework of Acceptability tool of data collection in the method section.

Authors’ Response:

Thanks for the suggestion. We have addressed this.

-regarding the sample size: could you please give some information about data saturation principle.

Authors’ Response:

Participants were recruited until the point of saturation (54) was reached. We determined that we had reached saturation when no new themes were being derived from subsequent interviews. It was possible to determine the saturation since the analytical process was an on-going process, taking place alongside the data collection. We also determined saturation by varying our participants and determining whether any new themes came up. 

2 you mention many limitations for the study that affect the validity of your results; what did you conduct to decrease the effect of them on your study results?

and you didn't ; why?

Authors’ Response:

The study had one notable limitation. Participants were initially screened for hearing loss using the Wulira App before being recruited for the FGDs. Of those screened, none was found to be having hearing loss. This might have influenced the participants’ optimism and perceptions, when being later asked to describe their experience and their perceived acceptability of the App. However, this potential bias was reduced by making participants not aware of their results before the FGDs and using open ended questions when conducting the FGDs.

Reviewer #2: General comments:

- Please avoid starting a sentence as "There is/was/are/were" in scientific writing.

Authors’ Response:

Thank you for this suggestion. We have addressed this.

- Do not use the full stop after the title of the section.

Authors’ Response:

Thank you for this observation. We have corrected this.

Abstract:

- Please say when and where the study was done.

Authors’ Response:

Thanks for the suggestion. We have addressed this. The study was conducted in a steel and iron manufacturing industry in Kampala during the month of April 2021.

- It is not clear which were the inclusion/exclusion criteria.

Authors’ Response:

Thank you for the observation. Due to the world limit, we had removed some of these aspects. The inclusion criteria included; permanent staff employed by the industry who were above 18 years and had consented to take part in the study. We excluded participants who had established history of hearing loss.

- It is not clear what was tested with the FGDs.

Authors’ Response:

FGDs explored group level perceptions regarding the Wulira app.

- The results are not specific (e.g., how many are considered friendly or cheap; what time-saving means, etc.)

Authors’ Response:

Qualitative results are not designed to be generalizable or to draw inference from the proportion within the sample that share a belief. As such, this information was not presented among our results.

Introduction:

- "has increased the risk of developing hearing loss" how considerable the risk is?

Authors’ Response:

The cited study done by Nondahl et al (5), on recreational firearm use revealed that, men (n=1538) who had ever regularly engaged in target shooting (odds ratio, 1.57; 95% confidence interval, 1.12-2.19) or who had done so in the past year (odds ratio, 2.00; 95% confidence interval, 1.15-3.46) were more likely to have a marked high-frequency hearing loss than those who had not. Risk of having a marked high-frequency hearing loss increased 7% for every 5 years (odds ratio, 1.07; 95% confidence interval, 1.03-1.12).

- Please also report the 95% confidence interval associated with Se and Sp.

Authors’ Response:

Thanks for the suggestion. Confidence intervals have been reported in the updated manuscript. Specificity of 93.2% (right ear, 95 % CI (88.1-95.4 %)) ,91.5% (left ear, 95 %CI (87.2-94.7)) and sensitivity of 91.4% (right ear, 95% CI (88.9-93.5%)), 88.4% (left ear, 95% CI (85.6-80.9)).

- "Previous studies" to which populations were these previously study conducted?

Authors’ Response:

Study by Meinke et al was carried out among industrial workers (manufacturing and administrative workers). The study population in this study was similar to ours. Study by Scheibe et al was carried out among diabetic patients aged 50 or older.

- "phone ownership is still low due to the cost" how low and how much does it cost relative to the income of persons who work in industries where hearing loss occurs?

Authors’ Response:

The Uganda National ICT Survey released in April 2018, revealed that 70.9% of individuals had a mobile phone. And of these, only 15.8% had smart phones. In terms of cost, most individuals on average spent UGX 14,500 per month on their phone. Among individuals that did not own a mobile phone, the cost of the mobile phone was the biggest barrier (cited by 88.9% of respondents), followed by the challenges of charging the phone battery (cited by 36.6% of respondents). 

https://www.nita.go.ug/sites/default/files/publications/National%20IT%20Survey%20April%2010th.pdf

- "Literature on the acceptability of mobile audiometry for monitoring purposes in Africa and Uganda is scarce" Briefly present these studies and their results.

Authors’ Response:

From literature review, studies that were found assessing hearing loss screening using mobile audiometry were among the pediatric population in South Africa and Malawi. Due to a different study population between our study and the reviewed studies, this statement has been removed from the updated manuscript.

- It is not clear which is hearing loss among industrial workers in the target population.

Authors’ Response:

The WHO defines disabling hearing loss as to hearing loss greater than 40dB in the better hearing ear in adults.

- End this section with the aim of your study.

Authors’ Response:

Thank you. We have included the aim of the study as suggested.

Methods:

- It is not clear when the study was conducted.

Authors’ Response:

Thanks for the observation. This has been updated accordingly. The study was conducted in a steel and iron manufacturing industry in Kampala during the month of April 2021.

- "industrial workers at a steel and iron manufacturing industry" Are the steel and iron manufacturing industry the most prevalent industry where hearing loss can occur?

Authors’ Response:

A study performed in Tanzania assessing industrial workers at risk of occupational hearing loss revealed that steel and iron factory workers had a prevalence of 48%. Tanzania having similar settings like Uganda, we felt it necessary to carry out our study in a similar study population. 

- Why was this particular city and industry was chosen?

Authors’ Response:

As of 2011, Kampala had 32% of the total number of manufacturing industries in Uganda. Steel and iron manufacturing industry employed the largest number of people in Kampala district which was close to 8,233 people. The selected industry employed over 1,100 staff on permanent basis thus having more than enough participants to meet the sample size of our study.

- The inclusion criteria are not sufficiently presented (e.g., how many years were they working in that industry, had the family history of hearing loss? etc.).

Authors’ Response:

Duration of work was not included in the inclusion but rather all permanent workers aged 18 years and above, and consented were included in the study.

- It is not clear how the participant was recruited and how the participants were decided.

Authors’ Response:

In selection of the participants, we discussed with the industry workers’ supervisor and selected those available, willing to participate, and working in a noisy section. Upon selection, they were then screened for hearing loss using the Wulira App. The screening was done by a trained audiologist who was part of the study team. After screening, the participants were then enrolled for the FGDs.

- Define "most active participants".

Authors’ Response:

“most active participants” were those that appeared to have more to say, but were not given enough time in the FGDs as observed by the moderator.

- Is there any reason why the participants were not evaluated for hearing loss by gold standard procedure?

Authors’ Response:

Wulira has already been validated against the gold standard and found to be as good (14). This current study was to explore acceptability of the Wulira App.

- It is not clear which questions were in the interviews.

Authors’ Response:

Thank you. The guide has been submitted as a supplementary to the manuscript. Some of the questions included;

a. In your opinion, what is occupational hearing loss? 

b. Have you undergone testing for hearing loss? If so, where did you go and what did they use to test you for hearing loss?

c. Was the Wulira app used to test your hearing ability? If it was used, how did you find the exercise? Would you mind telling me how the exercise went? (Affective attitude).

d. Tell me about the duration of the screening exercise using Wulira App. How long did it take to start and end? Was there any inconvenience you felt during this exercise? (Burden).

- It is not clear if the participants used the App on their own phones or not.

Authors’ Response:

The participants did not use the App on their phones but rather one of the study team member (FB) carried out this assessment using the App installed on the study tablet. This has been updated accordingly in the revised manuscript.

- It is not clear if the participants were or not with phone/smartphone skilled.

Authors’ Response:

The participants did not use the App on their phones but rather one of the study team member (FB) carried out hearing assessment using the Wulira App installed on the study tablet.

Results:

- Put the results either in text or in Tables. Do not duplicate the results in text and tables/figures.

Authors’ Response:

Thank you for the constructive feedback. We have improved the presentation of our results in text and those in table.

- Do not include in the body of the table the units of measurements (e.g. %).

Authors’ Response:

Thanks for this observation. This has been updated accordingly.

- "The results from the IDI and FGDs revealed themes that are presented using Sekhon’s framework to understand the acceptability of the hearing assessment using the Wulira app" this information is duplicated; please delete it.

Authors’ Response:

Thanks for the suggestion. This has been revised as suggested.

- It is not clear how many participants agreed with the features presented in Table 2 as facilitator/barrier.

Authors’ Response:

Qualitative results are not designed to be generalizable or to draw inference from the proportion within the sample that share a belief. As such, this information was not presented among our results.

Discussion:

- Do not duplicate information (the aim of the study is already known).

Authors’ Response:

Thanks for this observation. This has been revised accordingly.

- It is useful to begin the discussion by brieﬂy summarizing the main ﬁndings and explore possible mechanisms or explanations for these ﬁndings.

Authors’ Response:

Thank you for the suggestion. We have greatly improved the discussion of our results.

- Emphasize the new and important aspects of your study and put your findings in the context of the totality of the relevant evidence. State the limitations of your study, and explore the implications of your ﬁndings for future research and for clinical practice or policy.

Authors’ Response:

Thanks for the constructive feedback. We have greatly improved the discussion of our results.

- "Our results revealed that Wulira App’s compatibility with only smartphones will be a major barrier for individuals in rural areas or with no smart devices" a possible solution to this barrier is that the employer to provide the possibility to test personal hearing on a stand-alone smartphone on which the application is installed and why not with a supervision of a technology skilled person to assist individual persons.

Authors’ Response:

The authors appreciate the reviewer for this insightful feedback. The suggestion has been included in the revised manuscript.

- The main limitation of your study is the absence of performance evaluation of the used App since the respondent was disease-free and no standard gold diagnosis was made.

Authors’ Response:

Performance of the app was assessed elsewhere albeit in a different patient population. Wulira has already been validated against the gold standard and found to be as good (14). This current study was to explore acceptability of the Wulira App.

Conclusions:

- Not all sentences presented here are conclusions (e.g., 453-457).

Authors’ Response:

Thank you for the feedback. The conclusion has been updated to align with the results of the study.

- Please provide the date for ethics approval.

Authors’ Response:

The Uganda National Council for Science and Technology (UNCST) approved the research protocol on 22/02/2021. UNCST registration number is HS1237ES.

Reviewer #3: The reviewed article presents a qualitative evaluation of the acceptability of the Wulira App in assessing hearing loss among industrial workers in Kampala, Uganda.

Overall, this qualitative investigation produced a useful insight into both facilitating and hindering factors experienced by a sample of industrial workers after their hearing has been assessed using the evaluated App.

The evaluation involved 4 focus group discussions (FGD) with 8 participants per FGD and 12 in-depth interviews (IDI), conducted after all industrial workers were first tested for hearing loss using the evaluated App.

None of the investigated workers has been found to be suffering from hearing loss.

In this reviewer’s opinion, this might have influenced the participants’ optimism and perceptions, when being later asked to describe their experience and their perceived acceptability of the App.

Unless the audiometric test results were not communicated to participants until after their participation in the FGDs and IDIs (an aspect that should be clearly mentioned in the article), this represents a limitation that should also be mentioned by the authors, and which should be addressed by future studies, by including a larger number of participants which may ensure not only the qualitative-data saturation (which dictated sample size in this study), but also a reasonable number of participants with hearing problems to be detected by the App (thereby reflecting the prevalence of hearing loss in the targeted population of industrial workers).

Authors’ Response:

The authors thank the reviewer for this constructive feedback. This limitation has been included in the updated manuscript. Audiometric test results were not communicated to the participants prior to attending the FGDs. The results were communicated to them after the FGDs. This has been made clearer in the updated manuscript.

Despite the major conflict of interest that has been declared by 3 of the authors (who own, direct and/or have participated in the development of the evaluated App), other than this possibly important limitation of the study (which needs to be clarified or stated along with those already underlined in the Limitations section), the scientific approach appears to have been conducted using fairly objective methods (as described in the Methods part of the study and as reported in accordance to the COREQ guidelines).

Authors’ Response:

The authors thank the reviewer for this submission.

Also, lacking other evidence, the authors needed to rely on their own previous works when making accuracy/ sensitivity/ specificity claims about the investigated App. Given the aforementioned conflicts of interest, encouraging future independent studies of diagnostic accuracy and acceptability of the evaluated App would be desirable.

Authors’ Response:

The authors thank the reviewer for this suggestion.

Further improvements of the article manuscript may also be achieved by addressing the following issues:

Lines 73-75: please specify the gold standard against which the authors have determined the Se and Sp of the evaluated App in their previous work. Also, please specify the sample size and the 95% CI for Se and Sp.

Authors’ Response:

The gold standard was Pure Tone Audiometry (14). Confidence intervals have been reported in the updated manuscript. Specificity of 93.2% (right ear, 95 % CI (88.1-95.4 %)) ,91.5% (left ear, 95 %CI (87.2-94.7)) and sensitivity of 91.4% (right ear, 95% CI (88.9-93.5%)), 88.4% (left ear, 95% CI (85.6-80.9)).

Lines 129-130: please detail what “most active participants” means, since this may have been a highly subjective choice, leading to the selection of the most optimistic and favorably-oriented participants towards the evaluated App.

Authors’ Response:

“most active participants” were those that appeared to have more to say, but were not given enough time in the FGDs as observed by the moderator.

Lines 143-144: this reviewer suggests replacing “Before start of each FGD, moderator shared” with “Before the start of each FGD, the moderator shared”.

Thank you. The grammar in the statement has been rectified as suggested.

Line 150: this reviewer suggests replacing “moderator noted” with “the moderator noted”.

Thank you. The grammar has been rectified as suggested.

Line 172: this reviewer suggests replacing “FGD findings were ratified and triangulated themes got from IDI” with “The FGD findings were ratified and triangulated with themes got from IDI”.

Thank you. The grammar in the statement has been rectified as suggested.

Line 174: please specify which open code software was used to perform the analysis.

Thank you for this observation. Nvivo software was used to perform the analysis. This has been added in the updated manuscript.

Line 190: this reviewer suggests replacing “majority” with “the majority”.

Thank you. The grammar has been rectified as suggested.

Line 192: this reviewer suggests replacing “majority” with “a majority of them”.

Thank you. The grammar in the statement has been rectified as suggested.

Line 233: this reviewer suggests replacing “working high noise environments” with “working in high noise environments”.

Thank you. The grammar in the statement has been rectified as suggested

Line 235: there is an unlikely omission of a sense-giving verb in the verbatim transcription of Interviewee 10: “You never what result I would get.”

Thank you for this observation. The verb ‘Know’ was added in this verbatim transcription. 

Line 290: this reviewer suggests replacing “prevented some of them for accessing” with “prevented some of them from accessing”.

Thank you. The grammar in the statement has been rectified as suggested

Lines 382-383: this reviewer suggests replacing “ranges from everyone month to once a year” with “ranges from once a month to once a year”.

Thank you. The grammar in the statement has been rectified as suggested.

Line 398: this reviewer suggests replacing “Study done” with “A study done”.

Thank you. The grammar has been rectified as suggested.

Lines 401-402: Once again, please specify the gold standard against which the Se and Sp of the App and of PTA have been determined, in order to reach a conclusion of “similarity” between the App and conventional PTA. Also, even if Se and Sp were indeed similar between the App and conventional PTA, it is not clear how that would influence or even “explain its acceptability”.

The gold standard was Pure Tone Audiometry (14). Confidence intervals have been reported in the updated manuscript. Specificity of 93.2% (right ear, 95 % CI (88.1-95.4 %)) ,91.5% (left ear, 95 %CI (87.2-94.7)) and sensitivity of 91.4% (right ear, 95% CI (88.9-93.5%)), 88.4% (left ear, 95% CI (85.6-80.9)).

Lines 406-407: this reviewer suggests replacing “Study done” with “A study performed”.

Thank you. The grammar has been rectified as suggested.

Line 420: this reviewer suggests replacing “to separate individuals with potential hearing disorder from those who don’t” with “to separate individuals who have a potential hearing disorder from those who don’t”.

Thank you. The grammar in the statement has been rectified as suggested.

Line 424: this reviewer suggests replacing “in possession with these phones” with “in possession of these phones”.

Thank you. The grammar in the statement has been rectified as suggested.

Line 435: please resolve the doubling in “would would”.

Thank you. The repeated word has been deleted.

Lines 476-478 and 480: There are several abbreviations in the list which do not appear in the text of this article: YLD, OHL, NIHL, MDR-TB. Those abbreviations might have been useful in previous publications of the authors, but should be removed from this article.

Thank you for this observation. The abbreviation list has been updated accordingly.

---

## [Decision Letter · Decision Letter 1]

28 Feb 2022

PONE-D-21-32084R1ACCEPTABILITY OF THE WULIRA APP IN ASSESSING OCCUPATIONAL HEARING LOSS AMONG WORKERS IN A STEEL AND IRON MANUFACTURING INDUSTRY.PLOS ONE

Dear Dr. twinamasiko,

Thank you for submitting your manuscript to PLOS ONE. After careful consideration, we feel that it has merit but does not fully meet PLOS ONE’s publication criteria as it currently stands. Therefore, we invite you to submit a revised version of the manuscript that addresses the points raised during the review process.

We look forward to receiving your revised manuscript.

Kind regards,

Mohammad Hossein Ebrahimi

Academic Editor

PLOS ONE

**Comments to the Author**

1. Has the statistical analysis been performed appropriately and rigorously?

N/A

2. Have the authors made all data underlying the findings in their manuscript fully available?

No

3. Is the manuscript presented in an intelligible fashion and written in standard English?

No

Generally the authord appropriated response to my comments and suggestions. However, some chnages are needed before publication:

- I am not a native English speaker bu I belive that your manuscript could benefit by English language professional service.

- Write the aim of the study at past tense.

- Some duplicated text exist in the revised manuscript (e.g. Affective attitude).

- Line 230-231 belongs to the methods section.

- Do not start a sentence with a number.

- Please avoid starting a sentence as "There is/was/are/were" in scientific writing.

- Do not use the full stop after the title of the section.

Abstract:

- Please say when and where the study was done.

- It is not clear which were the inclusion/exclusion criteria.

- It is not clear what was tested with the FGDs.

- The results are not specific (e.g., how many are considered friendly or cheap; what time-saving means, etc.)

Introduction:

- "has increased the risk of developing hearing loss" how considerable the risk is?

- Please also report the 95% confidence interval associated with Se and Sp.

- "Previous studies" to which populations were these previously study conducted?

- "phone ownership is still low due to the cost" how low and how much does it cost relative to the income of persons who work in industries where hearing loss occurs?

- "Literature on the acceptability of mobile audiometry for monitoring purposes in Africa and Uganda is scarce" Briefly present these studies and their results.

- It is not clear which is hearing loss among industrial workers in the target population.

- End this section with the aim of your study.

Methods:

- It is not clear when the study was conducted.

- "industrial workers at a steel and iron manufacturing industry" Are the steel and iron manufacturing industry the most prevalent industry where hearing loss can occur?

- Why was this particular city and industry was chosen?

- The inclusion criteria are not sufficiently presented (e.g., how many years were they working in that industry, had the family history of hearing loss? etc.).

- It is not clear how the participant was recruited and how the participants were decided.

- Define "most active participants".

- Is there any reason why the participants were not evaluated for hearing loss by gold standard procedure?

- It is not clear which questions were in the interviews.

- It is not clear if the participants used the App on their own phones or not.

- It is not clear if the participants were or not with phone/smartphone skiled.

Results:

- Put the results either in text or in Tables. Do not duplicate the results in text and tables/figures.

- Do not include in the body of the table the units of measurements (e.g. %).

- "The results from the IDI and FGDs revealed themes that are presented using Sekhon’s framework to understand the acceptability of the hearing assessment using the Wulira app" this information is duplicated; please delete it.

- It is not clear how many participants agreed with the features presented in Table 2 as facilitator/barrier.

Discussion:

- Do not duplicate information (the aim of the study is already known).

- It is useful to begin the discussion by brieﬂy summarizing the main ﬁndings and explore possible mechanisms or explanations for these ﬁndings.

- Emphasize the new and important aspects of your study and put your findings in the context of the totality of the relevant evidence. State the limitations of your study, and explore the implications of your ﬁndings for future research and for clinical practice or policy.

- "Our results revealed that Wulira App’s compatibility with only smartphones will be a major barrier for individuals in rural areas or with no smart devices" a possible solution to this barrier is that the employer to provide the possibility to test personal hearing on a stand-alone smartphone on which the application is installed and why not with a supervision of a technology skilled person to assist individual persons.

- The main limitation of your study is the absence of performance evaluation of the used App since the respondent was disease-free and no standard gold diagnosis was made.

Conclusions:

- Not all sentences presented here are conclusions (e.g., 453-457).

- Please provide the date for ethics approval.

---

## [Author Response · Author response to Decision Letter 1]

13 Mar 2022

Generally, the authors appropriated response to my comments and suggestions. However, some changes are needed before publication:

- I am not a native English speaker but I believe that your manuscript could benefit by English language professional service.

Authors’ Response:

Thanks for the suggestion. The manuscript has been proof read and has been remarkably improved regarding the grammar.

- Write the aim of the study at past tense.

Authors’ Response:

Thanks for this observation. This has been updated as suggested.

- Some duplicated text exists in the revised manuscript (e.g. Affective attitude).

Authors’ Response:

Thanks for this observation. We have proof read the manuscript. The updated manuscript has no duplicated text.

- Line 230-231 belongs to the methods section.

Authors’ Response:

Lines 230-231 have statements in the result section. 

We believe you meant lines 440-441 in the limitation section. We acknowledge this statement should be in the methodology, however it was included here to explain the potential bias of our study and how we addressed this.

- Do not start a sentence with a number.

Authors’ Response:

Thanks for the suggestion. The sentence that was started with a number has been corrected as suggested.

Generally, the authors appropriated response to my comments and suggestions. However, some changes are needed before publication:

- I am not a native English speaker but I believe that your manuscript could benefit by English language professional service.

Authors’ Response:

Thanks for the suggestion. The manuscript has been proof read and has been remarkably improved regarding the grammar.

- Write the aim of the study at past tense.

Authors’ Response:

Thanks for this observation. This has been updated as suggested.

- Some duplicated text exists in the revised manuscript (e.g. Affective attitude).

Authors’ Response:

Thanks for this observation. We have proof read the manuscript. The updated manuscript has no duplicated text.

- Line 230-231 belongs to the methods section.

Authors’ Response:

Lines 230-231 have statements in the result section. 

We believe you meant lines 440-441 in the limitation section. We acknowledge this statement should be in the methodology, however it was included here to explain the potential bias of our study and how we addressed this.

- Do not start a sentence with a number.

Authors’ Response:

Thanks for the suggestion. The sentence that was started with a number has been corrected as suggested.

---

## [Editor Report · Decision Letter 2]

15 Mar 2022

PONE-D-21-32084R2ACCEPTABILITY OF THE WULIRA APP IN ASSESSING OCCUPATIONAL HEARING LOSS AMONG WORKERS IN A STEEL AND IRON MANUFACTURING INDUSTRY.PLOS ONE

Dear Dr. twinamasiko,

Thank you for submitting your manuscript to PLOS ONE. After careful consideration, we feel that it has merit but does not fully meet PLOS ONE’s publication criteria as it currently stands. Therefore, we invite you to submit a revised version of the manuscript that addresses the points raised during the review process.

We look forward to receiving your revised manuscript.

Kind regards,

Mohammad Hossein Ebrahimi

Academic Editor

PLOS ONE
---

## [Author Response · Author response to Decision Letter 2]

15 Mar 2022

Generally, the authors appropriated response to my comments and suggestions. However, some changes are needed before publication:

- I am not a native English speaker but I believe that your manuscript could benefit by English language professional service.

Authors’ Response:

Thanks for the suggestion. The manuscript has been proof read and has been remarkably improved regarding the grammar.

- Write the aim of the study at past tense.

Authors’ Response:

Thanks for this observation. This has been updated as suggested.

- Some duplicated text exists in the revised manuscript (e.g. Affective attitude).

Authors’ Response:

Thanks for this observation. We have proof read the manuscript. The updated manuscript has no duplicated text.

- Line 230-231 belongs to the methods section.

Authors’ Response:

Lines 230-231 have statements in the result section. 

We believe you meant lines 440-441 in the limitation section. We acknowledge this statement should be in the methodology, however it was included here to explain the potential bias of our study and how we addressed this.

- Do not start a sentence with a number.

Authors’ Response:

Thanks for the suggestion. The sentence that was started with a number has been corrected as suggested.

---

## [Editor Report · Decision Letter 3]

29 Mar 2022

ACCEPTABILITY OF THE WULIRA APP IN ASSESSING OCCUPATIONAL HEARING LOSS AMONG WORKERS IN A STEEL AND IRON MANUFACTURING INDUSTRY.

PONE-D-21-32084R3

Dear Dr. twinamasiko,

We’re pleased to inform you that your manuscript has been judged scientifically suitable for publication and will be formally accepted for publication once it meets all outstanding technical requirements.

Kind regards,

Mohammad Hossein Ebrahimi

Academic Editor

PLOS ONE
---

## [Editor Report · Acceptance letter]

22 Apr 2022

PONE-D-21-32084R3 

Acceptability of the Wulira app in assessing occupational hearing loss among workers in a steel and iron manufacturing industry 

Dear Dr. Twinamasiko:

I'm pleased to inform you that your manuscript has been deemed suitable for publication in PLOS ONE. Congratulations! Your manuscript is now with our production department. 

Kind regards, 

on behalf of

Dr. Mohammad Hossein Ebrahimi 

Academic Editor

PLOS ONE